# Meta-Gradient Reinforcement Learning

**Zhongwen Xu**
DeepMind
zhongwen@google.com

**Hado van Hasselt**
DeepMind
hado@google.com

**David Silver**
DeepMind
davidsilver@google.com

## Abstract

The goal of reinforcement learning algorithms is to estimate and/or optimise the value function. However, unlike supervised learning, no teacher or oracle is available to provide the true value function. Instead, the majority of reinforcement learning algorithms estimate and/or optimise a proxy for the value function. This proxy is typically based on a sampled and bootstrapped approximation to the true value function, known as a *return*. The particular choice of return is one of the chief components determining the nature of the algorithm: the rate at which future rewards are discounted; when and how values should be bootstrapped; or even the nature of the rewards themselves. It is well-known that these decisions are crucial to the overall success of RL algorithms. We discuss a gradient-based meta-learning algorithm that is able to adapt the nature of the return, online, whilst interacting and learning from the environment. When applied to 57 games on the Atari 2600 environment over 200 million frames, our algorithm achieved a new state-of-the-art performance.

The central goal of reinforcement learning (RL) is to optimise the agent's *return* (cumulative reward); this is typically achieved by a combination of prediction and control. The prediction subtask is to estimate the value function – the expected return from any given state. Ideally, this would be achieved by updating an approximate value function towards the true value function. The control subtask is to optimise the agent's policy for selecting actions, so as to maximise the value function. Ideally, the policy would simply be updated in the direction that increases the true value function. However, the true value function is unknown and therefore, for both prediction and control, a sampled return is instead used as a proxy. A large family of RL algorithms [Sutton, 1988, Rummery and Niranjan, 1994, van Seijen et al., 2009, Sutton and Barto, 2018], including several state-of-the-art deep RL algorithms [Mnih et al., 2015, van Hasselt et al., 2016, Harutyunyan et al., 2016, Hessel et al., 2018, Espeholt et al., 2018], are characterised by different choices of the return.

The *discount factor* $\gamma$ determines the time-scale of the return. A discount factor close to $\gamma = 1$ provides a long-sighted goal that accumulates rewards far into the future, while a discount factor close to $\gamma = 0$ provides a short-sighted goal that prioritises short-term rewards. Even in problems where long-sightedness is clearly desired, it is frequently observed that discounts $\gamma < 1$ achieve better results [Prokhorov and Wunsch, 1997], especially during early learning. It is known that many algorithms converge faster with lower discounts [Bertsekas and Tsitsiklis, 1996], but of course too low a discount can lead to highly sub-optimal policies that are too myopic. In practice it can be better to first optimise for a myopic horizon, e.g., with $\gamma = 0$ at first, and then to repeatedly increase the discount only after learning is somewhat successful [Prokhorov and Wunsch, 1997].

The return may also be *bootstrapped* at different time horizons. An *n-step return* accumulates rewards over $n$ time-steps and then adds the value function at the $n$th time-step. The $\lambda$-return [Sutton, 1988, Sutton and Barto, 2018] is a geometrically weighted combination of $n$-step returns. In either case, the meta-parameter $n$ or $\lambda$ can be important to the performance of the algorithm, trading off bias and variance. Many researchers have sought to automate the selection of these parameters [Kearns and Singh, 2000, Downey and Sanner, 2010, Konidaris et al., 2011, White and White, 2016].

There are potentially many other design choices that may be represented in the return, including off-policy corrections [Espeholt et al., 2018, Munos et al., 2016], target networks [Mnih et al., 2015], emphasis on certain states [Sutton et al., 2016], reward clipping [Mnih et al., 2013], or even the nature of the rewards themselves [Randløv and Alstrøm, 1998, Singh et al., 2005, Zheng et al., 2018].

In this work, we are interested in one of the fundamental problems in reinforcement learning: what would be the best form of return for the agent to maximise? Specifically, we propose to *learn* the return function by treating it as a parametric function with tunable meta-parameters $\eta$, for instance including the discount factor $\gamma$, or the bootstrapping parameter $\lambda$ [Sutton, 1988]. The meta-parameters $\eta$ are adjusted *online* during the agent's interaction with the environment, allowing the return to both adapt to the specific problem, and also to dynamically adapt over time to the changing context of learning. We derive a practical gradient-based meta-learning algorithm and show that this can significantly improve performance on large-scale deep reinforcement learning applications.

# 1 Meta-Gradient Reinforcement Learning Algorithms

In deep reinforcement learning, the value function and policy are approximated by a neural network with parameters $\theta$, denoted by $v_\theta(S)$ and $\pi_\theta(A|S)$ respectively. At the core of the algorithm is an *update function*,

$$\theta' = \theta + f(\tau, \theta, \eta)\,, \tag{1}$$

that adjusts parameters from a sequence of experience $\tau_t = \{S_t, A_t, R_{t+1}, \ldots\}$ consisting of states $S$, actions $A$ and rewards $R$. The nature of the function is determined by *meta-parameters* $\eta$.

Our meta-gradient RL approach is based on the principle of online cross-validation [Sutton, 1992], using successive samples of experience. The underlying RL algorithm is applied to the first sample (or samples), and its performance is measured in a subsequent sample. Specifically, the algorithm starts with parameters $\theta$, and applies the update function to the first sample(s), resulting in new parameters $\theta'$. The gradient $\mathrm{d}\theta'/\mathrm{d}\eta$ of these updates indicates how the meta-parameters affected these new parameters.

The algorithm then measures the performance of the new parameters $\theta'$ on a second sample $\tau'$. For instance, when learning online $\tau'$ could be the next time-step immediately following $\tau$. Performance is measured by a differentiable *meta-objective* $\bar{J}(\tau', \theta', \bar{\eta})$ that uses a fixed meta-parameter $\bar{\eta}$.

The gradient of the meta-objective with respect to the meta-parameters $\eta$ is obtained by applying the chain rule:

$$\frac{\partial \bar{J}(\tau', \theta', \bar{\eta})}{\partial \eta} = \frac{\partial \bar{J}(\tau', \theta', \bar{\eta})}{\partial \theta'} \frac{\mathrm{d}\theta'}{\mathrm{d}\eta}\,. \tag{2}$$

To compute the gradient of the updates, $\mathrm{d}\theta'/\mathrm{d}\eta$, we note that the parameters form an additive sequence, and the gradient can therefore be accumulated online [Williams and Zipser, 1989],

$$\frac{\mathrm{d}\theta'}{\mathrm{d}\eta} = \frac{\mathrm{d}\theta}{\mathrm{d}\eta} + \frac{\partial f(\tau, \theta, \eta)}{\partial \eta} + \frac{\partial f(\tau, \theta, \eta)}{\partial \theta}\frac{\mathrm{d}\theta}{\mathrm{d}\eta} = \left(\mathrm{I} + \frac{\partial f(\tau, \theta, \eta)}{\partial \theta}\right)\frac{\mathrm{d}\theta}{\mathrm{d}\eta} + \frac{\partial f(\tau, \theta, \eta)}{\partial \eta} \tag{3}$$

This update has the form

$$z' = \mathrm{A}z + \frac{\partial f(\tau, \theta, \eta)}{\partial \eta}\,,$$

where $z = \mathrm{d}\theta/\mathrm{d}\eta$ and $z' = \mathrm{d}\theta'/\mathrm{d}\eta$.

The exact gradient is given by $\mathrm{A} = \mathrm{I} + \partial f(\tau, \theta, \eta)/\partial \theta$. In practice, the gradient $\partial f(\tau, \theta, \eta)/\partial \theta$ is large and challenging to compute — it is a $n \times n$ matrix, where $n$ is the number of parameters in $\theta$. In practice, we approximate the gradient, $z \approx \mathrm{d}\theta/\mathrm{d}\eta$. One possibility is to use an alternate update $\mathrm{A} = \mathrm{I} + \hat{\partial} f(\tau, \theta, \eta)/\hat{\partial}\theta$ using a cheap approximate derivative $\hat{\partial} f(\tau, \theta, \eta)/\hat{\partial}\theta \approx \partial f(\tau, \theta, \eta)/\partial \theta$, for instance using a diagonal approximation [Sutton, 1992, Schraudolph, 1999]. Furthermore, the gradient accumulation defined above assumes that the meta-parameters $\eta$ are held fixed throughout training. In practice, we are updating $\eta$ and therefore it may be desirable to decay the trace into the past [Schraudolph, 1999], $\mathrm{A} = \mu(\mathrm{I} + \partial f(\tau, \theta, \eta)/\partial \theta)$, using decay rate $\mu \in [0, 1]$. The simplest approximation is to use $\mathrm{A} = 0$ (or equivalently $\mu = 0$), which means that we only consider the effect of the meta-parameters $\eta$ on a single update; this approximation is especially cheap to compute.

Finally, the meta-parameters $\eta$ are updated to optimise the meta-objective, for example by applying stochastic gradient descent (SGD) to update $\eta$ in the direction of the meta-gradient,

$$\Delta\eta = -\beta\frac{\partial\bar{J}(\tau',\theta',\bar{\eta})}{\partial\theta'}z', \tag{4}$$

where $\beta$ is the learning rate for updating meta parameter $\eta$. The pseudo-code for the meta-gradient reinforcement learning algorithm is provided in Appendix A.

In the following sections we instantiate this idea more specifically to RL algorithms based on predicting or controlling returns. We begin with a pedagogical example of using meta-gradients for prediction using a temporal-difference update. We then consider meta-gradients for control, using a canonical actor-critic update function and a policy gradient meta-objective. Many other instantiations of meta-gradient RL would be possible, since the majority of deep reinforcement learning updates are differentiable functions of the return, including, for instance, value-based methods like SARSA($\lambda$) [Rummery and Niranjan, 1994, Sutton and Barto, 2018] and DQN [Mnih et al., 2015], policy-gradient methods [Williams, 1992], or actor-critic algorithms like A3C [Mnih et al., 2016] and IMPALA [Espeholt et al., 2018].

### 1.1 Applying Meta-Gradients to Returns

We define the return $g_\eta(\tau_t)$ to be a function of an episode or a truncated $n$-step sequence of experience $\tau_t = \{S_t, A_t, R_{t+1}, \dots, S_{t+n}\}$. The nature of the return is determined by the meta-parameters $\eta$.

The $n$-step return [Sutton and Barto, 2018] accumulates rewards over the sequence and then bootstraps from the value function,

$$g_\eta(\tau_t) = R_{t+1} + \gamma R_{t+2} + \gamma^2 R_{t+3} + \dots, +\gamma^{n-1}R_{t+n} + \gamma^n v_\theta(S_{t+n}) \tag{5}$$

where $\eta = \{\gamma\}$.

The $\lambda$-return is a geometric mixture of $n$-step returns, [Sutton, 1988]

$$g_\eta(\tau_t) = R_{t+1} + \gamma(1-\lambda)v_\theta(S_{t+1}) + \gamma\lambda g_\eta(\tau_{t+1}) \tag{6}$$

where $\eta = \{\gamma, \lambda\}$. The $\lambda$-return has the advantage of being fully differentiable with respect to the meta-parameters. The meta-parameters $\eta$ may be viewed as gates that cause the return to terminate ($\gamma = 0$) or bootstrap ($\lambda = 0$), or to continue onto the next step ($\gamma = 1$ and $\lambda = 1$). The $n$-step or $\lambda$-return can be augmented with off-policy corrections [Precup et al., 2000, Sutton et al., 2014, Espeholt et al., 2018] if it is necessary to correct for the distribution used to generate the data.

A typical RL algorithm would hand-select the meta-parameters, such as the discount factor $\gamma$ and bootstrapping parameter $\lambda$, and these would be held fixed throughout training. Instead, we view the return $g$ as a function parameterised by meta-parameters $\eta$, which may be differentiated to understand its dependence on $\eta$. This in turn allows us to compute the gradient $\partial f/\partial\eta$ of the update function with respect to the meta-parameters $\eta$, and hence the meta-gradient $\partial\bar{J}(\tau',\theta',\bar{\eta})/\partial\eta$. In essence, our agent asks itself the question, "which return results in the best performance?", and adjusts its meta-parameters accordingly.

### 1.2 Meta-Gradient Prediction

We begin with a simple instantiation of the idea, based on the canonical TD($\lambda$) algorithm for prediction. The objective of the TD($\lambda$) algorithm (according to the forward view [Sutton and Barto, 2018]) is to minimise the squared error between the value function approximator $v_\theta(S)$ and the $\lambda$-return $g_\eta(\tau)$,

$$J(\tau,\theta,\eta) = (g_\eta(\tau) - v_\theta(S))^2 \qquad \frac{\partial J(\tau,\theta,\eta)}{\partial\theta} = -2(g_\eta(\tau) - v_\theta(S))\frac{\partial v_\theta(S)}{\partial\theta} \tag{7}$$

where $\tau$ is a sampled trajectory starting with state $S$, and $\partial J(\tau,\theta,\eta)/\partial\theta$ is a semi-gradient [Sutton and Barto, 2018], i.e. the $\lambda$-return is treated as constant with respect to $\theta$.

The TD($\lambda$) update function $f(\cdot)$ applies SGD to update the agent's parameters $\theta$ to descend the gradient of the objective with respect to the parameters,

$$f(\tau,\theta,\eta) = -\frac{\alpha}{2}\frac{\partial J(\tau,\theta,\eta)}{\partial\theta} = \alpha(g_\eta(\tau) - v_\theta(S))\frac{\partial v_\theta(S)}{\partial\theta} \tag{8}$$

where $\alpha$ is the learning rate for updating agent $\theta$. We note that this update is itself a differentiable function of the meta-parameters $\eta$,

$$\frac{\partial f(\tau, \theta, \eta)}{\partial \eta} = -\frac{\alpha}{2}\frac{\partial^2 J(\tau, \theta, \eta)}{\partial \theta \, \partial \eta} = \alpha \frac{\partial g_\eta(\tau)}{\partial \eta}\frac{\partial v_\theta(S)}{\partial \theta} \tag{9}$$

The key idea of the meta-gradient prediction algorithm is to adjust meta-parameters $\eta$ in the direction that achieves the best predictive accuracy. This is measured by cross-validating the new parameters $\theta'$ on a second trajectory $\tau'$ that starts from state $S'$, using a mean squared error (MSE) meta-objective and taking its semi-gradient,

$$\bar{J}(\tau', \theta', \bar{\eta}) = (g_{\bar{\eta}}(\tau') - v_{\theta'}(S'))^2 \qquad \frac{\partial \bar{J}(\tau', \theta', \bar{\eta})}{\partial \theta'} = -2(g_{\bar{\eta}}(\tau') - v_{\theta'}(S'))\frac{\partial v_{\theta'}(S')}{\partial \theta'} \tag{10}$$

The meta-objective in this case could make use of an unbiased and long-sighted return[1], for example using $\bar{\eta} = \{\bar{\gamma}, \bar{\lambda}\}$ where $\bar{\gamma} = 1$ and $\bar{\lambda} = 1$.

### 1.3 Meta-Gradient Control

We now provide a practical example of meta-gradients applied to control. We focus on the A2C algorithm – an actor-critic update function that combines both prediction and control into a single update. This update function is widely used in several state-of-the-art agents [Mnih et al., 2016, Jaderberg et al., 2017b, Espeholt et al., 2018]. The semi-gradient of the A2C objective, $\partial J(\tau; \theta, \eta)/\partial \theta$, is defined as follows,

$$-\frac{\partial J(\tau, \theta, \eta)}{\partial \theta} = (g_\eta(\tau) - v_\theta(S))\frac{\partial \log \pi_\theta(A|S)}{\partial \theta} + b(g_\eta(\tau) - v_\theta(S))\frac{\partial v_\theta(S)}{\partial \theta} + c\frac{\partial H(\pi_\theta(\cdot|S))}{\partial \theta}. \tag{11}$$

The first term represents a control objective, encouraging the policy $\pi_\theta$ to select actions that maximise the return. The second term represents a prediction objective, encouraging the value function approximator $v_\theta$ to more accurately estimate the return $g_\eta(\tau)$. The third term regularises the policy according to its entropy $H(\pi_\theta)$, and $b$, $c$ are scalar coefficients that weight the different components in the objective function.

The A2C update function $f(\cdot)$ applies SGD to update the agent's parameters $\theta$. This update function is a differentiable function of the meta-parameters $\eta$,

$$f(\tau, \theta, \eta) = -\alpha\frac{\partial J(\tau, \theta, \eta)}{\partial \theta} \qquad \frac{\partial f(\tau, \theta, \eta)}{\partial \eta} = \alpha\frac{\partial g_\eta(\tau)}{\partial \eta}\left[\frac{\partial \log \pi_\theta(A|S)}{\partial \theta} + b\frac{\partial v_\theta(S)}{\partial \theta}\right] \tag{12}$$

Now we come to the choice of meta-objective $\bar{J}$ to use for control. Our goal is to identify the return function that maximises overall performance in our agents. This may be directly measured by a meta-objective focused exclusively on optimising returns – in other words a policy gradient objective,

$$\frac{\partial \bar{J}(\tau', \theta', \bar{\eta})}{\partial \theta'} = (g_{\bar{\eta}}(\tau') - v_{\theta'}(S'))\frac{\partial \log \pi_{\theta'}(A'|S')}{\partial \theta'}. \tag{13}$$

This equation evaluates how good the updated policy $\theta'$ is in terms of returns computed under $\bar{\eta}$, when measured on "held-out" experiences $\tau'$, e.g. the subsequent $n$-step trajectory. When cross-validating performance using this meta-objective, we use fixed meta-parameters $\bar{\eta}$, ideally representing a good proxy to the true objective of the agent. In practice this typically means selecting reasonable values of $\bar{\eta}$; the agent is free to adapt its meta-parameters $\eta$ and choose values that perform better in practice.

We now put the meta-gradient control algorithm together. First, the parameters $\theta$ are updated on a sample of experience $\tau$ using the A2C update function (Equation (11)), and the gradient of the update (Equation (12)) is accumulated into trace $z$. Second, the performance is cross-validated on a subsequent sample of experience $\tau'$ using the policy gradient meta-objective (Equation (13)). Finally, the meta-parameters $\eta$ are updated according to the gradient of the meta-objective (Equation (4)).

## 1.4 Conditioned Value and Policy Functions

One complication of the approach outlined above is that the return function $g_\eta(\tau)$ is non-stationary, adapting along with the meta-parameters throughout the training process. As a result, there is a danger that the value function $v_\theta$ becomes inaccurate, since it may be approximating old returns. For example, the value function may initially form a good approximation of a short-sighted return with $\gamma = 0$, but if $\gamma$ subsequently adapts to $\gamma = 1$ then the value function may suddenly find its approximation is rather poor. The same principle applies for the policy $\pi$, which again may have specialised to old returns.

To deal with non-stationarity in the value function and policy, we utilise an idea similar to universal value function approximation (UVFA) [Schaul et al., 2015]. The key idea is to provide the meta-parameters $\eta$ as an additional input to condition the value function and policy, as follows:

$$v_\theta^\eta(S) = v_\theta([S; \mathbf{e}_\eta]), \qquad\qquad \pi_\theta^\eta(S) = \pi_\theta([S; \mathbf{e}_\eta]),$$

where $\mathbf{e}_\eta$ is the embedding of $\eta$, $[s; \mathbf{e}_\eta]$ denotes concatenation of vectors $s$ and $\mathbf{e}_\eta$, the embedding network $\mathbf{e}_\eta$ is updated by backpropagation during training but the gradient is not flowing through $\eta$.

In this way, the agent explicitly learns value functions and policies that are appropriate for various $\eta$. The approximation problem becomes a little harder, but the payoff is that the algorithm can freely shift the meta-parameters without needing to wait for the approximator to "catch up".

## 1.5 Meta-Gradient Reinforcement Learning in Practice

To scale up the meta-gradient approach, several additional steps were taken. For efficiency, the A2C objective and meta-objective were accumulated over all time-steps within an $n$-step trajectory of experience. The A2C objective was optimised by RMSProp [Tieleman and Hinton, 2012] without momentum [Mnih et al., 2015, 2016, Espeholt et al., 2018]. This is a differentiable function of the meta-parameters, and can therefore be substituted similarly to SGD (see Equation (12)); this process may be simplified by automatic differentiation (Appendix C.2). As in IMPALA, an off-policy correction was used, based on a V-trace return (see Appendix C.1). For efficient implementation, mini-batches of trajectories were computed in parallel; trajectories were reused twice for both the update function and for cross-validation (see Appendix C.3).

## 2 Illustrative Examples

To illustrate the key idea of our meta-gradient approach, we provide two examples that show how the discount factor $\gamma$ and temporal difference parameter $\lambda$, respectively, can be meta-learned. We focus on meta-gradient prediction using the TD($\lambda$) algorithm and a MSE meta-objective with $\bar{\gamma} = 1$ and $\bar{\lambda} = 1$, as described in Section 1.2. For these illustrative examples, we consider *state-dependent* meta-parameters that can take on a different value in each state.

The first example is a 10-step Markov reward process (MRP), that alternates between "signal" and "noise" transitions. Transitions from odd-numbered "signal" states receive a small positive reward, $R = +0.1$. Transitions from even-numbered "noise" states receive a random reward, $R \sim \mathcal{N}(0, 1)$. To ensure that the signal can overwhelm the noise, it is beneficial to terminate the return (low $\gamma$) in "noise" states, but to continue the return (high $\gamma$) in "signal" states.

The second example is a 9-step MRP, that alternates between random rewards and the negation of whatever reward was received on the previous step. The sum of rewards over each such pair of time-steps is zero. There are 9 transitions, so the last reward is always random. To predict accurately, it is beneficial to bootstrap (low $\lambda$) in states for which the value function is well-known and equal to zero, but to avoid bootstrapping (high $\lambda$) in the noisier, partially observed state for which the return will depend on the previous reward, which cannot be inferred from the state itself.

Figure 1 shows the results of meta-gradient prediction using the TD($\lambda$) algorithm. The meta-gradient algorithm was able to adapt both $\lambda$ and $\gamma$ to form returns that alternate between high or low values in odd or even states respectively.

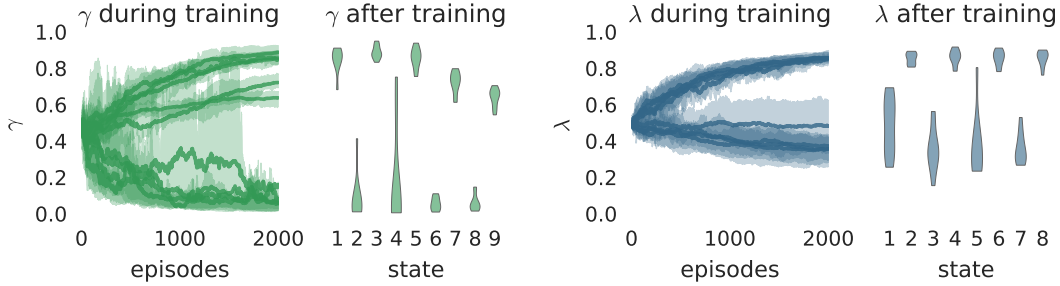

(a) Chain MRP. For the adaptive $\gamma$ experiments, the rewards alternate between $+0.1$ and zero-mean Gaussian on each step. For the adaptive $\lambda$ experiments, the rewards alternate between zero-mean Gaussians and the negative of whatever reward was received on the previous step: $R_{t+1} = -R_t$.

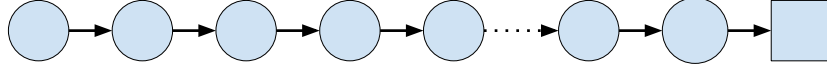

Figure 1: Illustrative results of meta-gradient learning of a state-dependent (a) bootstrapping parameter $\lambda$ or (b) discount factor $\gamma$, in the respective Markov reward processes (top). In each of the subplot shown in the bottom, the first one shows how the meta-parameter $\gamma$ or $\lambda$ adapts over the course of training (averaged over 10 seeds - shaded regions cover 20%–80% percentiles). The second plot shows the final value of $\gamma$ or $\lambda$ in each state, identifying appropriately high/low values for odd/even states respectively (violin plots show distribution over seeds).

## 3   Deep Reinforcement Learning Experiments

In this section, we demonstrate the advantages of the proposed meta-gradient learning approach using a state-of-the-art actor-critic framework IMPALA [Espeholt et al., 2018]. We focused on adapting the discount factor $\eta = \{\gamma\}$ (which we found to be the most effective meta-parameter in preliminary experiments). We also investigated adapting the bootstrapping parameter $\lambda$. For these experiments, the meta-parameters were state-independent, adapting one scalar value for $\gamma$ and $\lambda$ respectively (state-dependent meta-parameters did not provide significant benefit in preliminary experiments).[2]

### 3.1   Experiment Setup

We validate the proposed approach on Atari 2600 video games from Arcade Learning Environment (ALE) [Bellemare et al., 2013], a standard benchmark for deep reinforcement learning algorithms. We build our agent with the IMPALA framework [Espeholt et al., 2018], an efficient distributed implementation of actor-critic architecture [Sutton and Barto, 2018, Mnih et al., 2016]. We utilise the deep ResNet architecture [He et al., 2016] specified in Espeholt et al. [2018], which has shown great advantages over the shallow architecture [Mnih et al., 2015]. Following Espeholt et al., we train our agent for 200 million frames. Our algorithm does not require extra data compared to the baseline algorithms, as each experience can be utilised in both training the agent itself and training the meta parameters $\eta$ (i.e., each experience can serve as validation data of other experiences). We describe the detailed implementation in the Appendix C.3. For full details about the IMPALA implementation and the specific off-policy correction $g_\eta(\tau)$, please refer to Espeholt et al. [2018].

The agents are evaluated on 57 different Atari games and the median of human-normalised scores [Nair et al., 2015, van Hasselt et al., 2016, Wang et al., 2016b, Mnih et al., 2016] are reported. There are two different evaluation protocols. The first protocol is "human starts" [Nair et al., 2015, Wang et al., 2016b, van Hasselt et al., 2016], which initialises episodes to a state that is randomly sampled from human play. The second protocol is "no-ops starts", which initialises each episode with a random sequence of no-op actions; this protocol is also used during training. We keep the configuration (e.g., batch size, unroll length, learning rate, entropy cost) the same as specified in Espeholt et al. [2018] for a fair comparison. For self-contained purpose, we provide all of the important hyper-parameters used

|  | $\eta$ | Human starts | | No-op starts | |
|---|---|---|---|---|---|
|  |  | $\gamma = 0.99$ | $\gamma = 0.995$ | $\gamma = 0.99$ | $\gamma = 0.995$ |
| IMPALA | $\{\}$ | 144.4% | 211.9% | 191.8% | 257.1% |
| Meta-gradient | $\{\lambda\}$ | 156.6% | 214.2% | 185.5% | 246.5% |
|  |  | $\bar{\gamma} = 0.99$ | $\bar{\gamma} = 0.995$ | $\bar{\gamma} = 0.99$ | $\bar{\gamma} = 0.995$ |
| Meta-gradient | $\{\gamma\}$ | 233.2% | 267.9% | 280.9% | 275.5% |
| Meta-gradient | $\{\gamma, \lambda\}$ | 221.6% | 292.9% | 242.6% | 287.6% |

Table 1: Results of meta-learning the discount parameter $\gamma$, the temporal-difference learning parameter $\lambda$, or both $\gamma$ and $\lambda$, compared to the baseline IMPALA algorithm which meta-learns neither. Results are given both for the discount factor $\gamma = 0.99$ originally reported in [Espeholt et al., 2018] and also for a tuned discount factor $\gamma = 0.995$ (see Appendix D.1); the cross-validated discount factor $\bar{\gamma}$ in the meta-objective was set to the same value for a fair comparison.

in this paper, including the ones following Espeholt et al. [2018] and the additional meta-learning optimisation hyper-parameters (i.e., meta batch size, meta learning rate $\beta$, embedding size for $\eta$), in Appendix B. The meta-learning hyper-parameters are chosen according to the performance of six Atari games as common practice in Deep RL Atari experiments [van Hasselt et al., 2016, Mnih et al., 2016, Wang et al., 2016b]. Additional implementation details are provided in Appendix C.

## 3.2 Experiment Results

We compared four variants of the IMPALA algorithm: the original baseline algorithm without meta-gradients, i.e. $\eta = \{\}$; using meta-gradients with $\eta = \{\lambda\}$; using meta-gradients with $\eta = \{\gamma\}$; and using meta-gradients with $\eta = \{\gamma, \lambda\}$. The original IMPALA algorithm used a discount factor of $\gamma = 0.99$; however, when we manually tuned the discount factor and found that a discount factor of $\gamma = 0.995$ performed considerably better (see Appendix D.1). For a fair comparison, we tested our meta-gradient algorithm in both cases. When the discount factor is not adapted, $\eta = \{\}$ or $\eta = \{\lambda\}$, we used a fixed value of $\gamma = 0.99$ or $\gamma = 0.995$. When the discount factor is adapted, $\eta = \{\gamma\}$ or $\eta = \{\gamma, \lambda\}$, we cross-validate with a meta-parameter of $\bar{\gamma} = 0.99$ or $\bar{\gamma} = 0.995$ accordingly in the meta-objective $\bar{J}$ (Equation (13)). Manual tuning of the $\lambda$ parameter did not have a significant impact on performance and we therefore compared only to the original value of $\lambda = 1$.

We summarise the median human-normalised scores in Table 1; individual improvements on each game, compared to the IMPALA baseline, are given in Appendix E.1; and individual plots demonstrating the adaptation of $\gamma$ and $\lambda$ are provided in Appendix E.2. The meta-gradient RL algorithm increased the median performance, compared to the baseline algorithm, by a margin between 30% and 80% across "human starts" and "no-op starts" conditions, and with both $\gamma = 0.99$ and $\gamma = 0.995$.

We also verified the architecture choice of conditioning the value function $v$ and policy $\pi$ on the meta-parameters $\eta$. We compared the proposed algorithm with an identical meta-gradient algorithm that adapts the discount factor $\eta = \{\gamma\}$, but does not provide an embedding of the discount factor as an input to $\pi$ and $v$. For this experiment, we used a cross-validation discount factor of $\bar{\gamma} = 0.995$. The human-normalised median score was only 183%, well below the IMPALA baseline with $\gamma = 0.995$ (211.9%), and much worse than the full meta-gradient algorithm that includes the discount factor embedding (267.9%).

Finally, we compare against the state-of-the-art agent trained on Atari games, namely Rainbow [Hessel et al., 2018], which combines DQN [Mnih et al., 2015] with double Q-learning [van Hasselt et al., 2016, van Hasselt, 2010], prioritised replay [Schaul et al., 2016], dueling networks [Wang et al., 2016b], multi-step targets [Sutton, 1988, Sutton and Barto, 2018], distributional RL [Bellemare et al., 2017], and parameter noise for exploration [Fortunato et al., 2018]. Rainbow obtains median human-normalised score of 153% on the human starts protocol and 223% on the no-ops protocol. In contrast, the meta-gradient agent achieved a median score of 292.9% on human starts and 287.6% on no-ops, with the same number (200M) of frames. We note, however, that there are many differences between the two algorithms, including the deeper neural network architecture used in our work.

# 4 Related Work

Among the earliest studies on meta learning (or learning to learn [Thrun and Pratt, 1998]), Schmidhuber [1987] applied genetic programming to itself to evolve better genetic programming algorithms. Hochreiter et al. [2001] used recurrent neural networks like Long Short-Term Memory (LSTM) [Hochreiter and Schmidhuber, 1997] as meta-learners. A recent direction of research has been to meta-learn an *optimiser* using a recurrent parameterisation [Andrychowicz et al., 2016, Wichrowska et al., 2017]. Duan et al. [2016] and Wang et al. [2016a] proposed to learn a recurrent meta-policy that itself learns to solve the reinforcement learning problem, so that the recurrent policy can generalise into new tasks faster than learning the policy from scratch. Model-Agnostic Meta-Learning (MAML) [Finn et al., 2017a, Finn and Levine, 2018, Finn et al., 2017b, Grant et al., 2018, Al-Shedivat et al., 2018] learns a good initialisation of the model that can adapt quickly to other tasks within a few gradient update steps. These works focus on a multi-task setting in which meta-learning takes place on a distribution of training tasks, to facilitate fast adaptation on an unseen test task. In contrast, our work emphasises the (arguably) more fundamental problem of meta-learning within a single task. In other words we return to the standard formulation of RL as maximising rewards during a single lifetime of interactions with an environment.

Contemporaneously with our own work, Zheng et al. [2018] also propose a similar algorithm to learn meta-parameters of the return: in their case an auxiliary reward function that is added to the external rewards. They do not condition their value function or policy, and reuse the same samples for both the update function and the cross-validation step – which may be problematic in stochastic domains when the noise these updates becomes highly correlated.

There are many works focusing on adapting learning rate through gradient-based methods [Sutton, 1992, Schraudolph, 1999, Maclaurin et al., 2015, Pedregosa, 2016, Franceschi et al., 2017], Bayesian optimisation methods [Snoek et al., 2012], or evolution based hyper-parameter tuning [Jaderberg et al., 2017a, Elfwing et al., 2017]. In particular, Sutton [1992], introduced the idea of online cross-validation; however, this method was limited in scope to adapting the learning rate for linear updates in supervised learning (later extended to non-linear updates by Schraudolph [1999]); whereas we focus on the fundamental problem of reinforcement learning, i.e., adapting the return function to maximise the proxy returns we can achieve from the environment.

There has also been significant prior work on automatically adapting the bootstrapping parameter $\lambda$. Singh and Dayan [1998] empirically analyse the effect of $\lambda$ in terms of bias, variance and MSE. Kearns and Singh [2000] derive upper bounds on the error of temporal-difference algorithms, and use these bounds to derive schedules for $\lambda$. Downey and Sanner [2010] introduced a Bayesian model averaging approach to scheduling $\lambda$. Konidaris et al. [2011] derive a maximum-likelihood estimator, TD($\gamma$), that weights the $n$-step returns according to the discount factor, leading to a parameter-free algorithm for temporal-difference learning with linear function approximation. White and White [2016] introduce an algorithm that explicitly estimates the bias and variance, and greedily adapts $\lambda$ to locally minimise the MSE of the $\lambda$-return. Unlike our meta-gradient approach, these prior approaches exploit i.i.d. assumptions on the trajectory of experience that are not realistic in many applications.

# 5 Conclusion

In this work, we discussed how to learn the meta-parameters of a return function. Our meta-learning algorithm runs online, while interacting with a single environment, and successfully adapts the return to produce better performance. We demonstrated, by adjusting the meta-parameters of a state-of-the-art deep learning algorithm, that we could achieve much higher performance than previously observed on 57 Atari 2600 games from the Arcade Learning Environment.

Our proposed method is more general, and can be applied not just to the discount factor or bootstrapping parameter, but also to other components of the return, and even more generally to the learning update itself. Hyper-parameter tuning has been a thorn in the side of reinforcement learning research for several decades. Our hope is that this approach will allow agents to automatically tune their own hyper-parameters, by exposing them as meta-parameters of the learning update. This may also result in better performance because the parameters can change over time and adapt to novel environments.

## Acknowledgements

The authors would like to thank Matteo Hessel, Lasse Espeholt, Hubert Soyer, Dan Horgan, Aedan Pope and Tim Harley for their kind engineering support; and Joseph Modayil, Andre Barreto for their suggestions and comments on an early version of the paper. The authors would also like to thank anonymous reviewers for their constructive suggestions on improving the paper.

## Footnotes

[1]The meta-objective could even use a discount factor that is longer-sighted than the original problem, perhaps spanning over many episodes.

[2]In practice we parameterise $\eta = \sigma(x)$, where $\sigma$ is the logistic function $\sigma(x) = \frac{1}{1+e^{-x}}$; i.e. the meta-parameters are actually the logits of $\gamma$ and $\lambda$.

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
