[Supplementary Material]

# A Pseudo-code for Meta-Gradient Reinforcement Learning Algorithms

In this section, we provide the pseudo-code for meta-gradient reinforcement learning algorithms:

---

**Algorithm 1:** Meta-Gradient Reinforcement Learning

---

**Input:** minibatch size $B$, meta minibatch size $B'$, learning rate $\alpha$, meta learning rate $\beta$ and reference meta-parameter $\bar{\eta}$ .

1 Initialise agent parameter $\theta$ and meta-parameter $\eta$ ;
2 **while** *training* **do**
3      Sample trajectories $\mathcal{T} = \{\tau_1, \tau_2, \ldots, \tau_B\}$ for updating agent $\theta$ ;
4      Sample trajectories $\mathcal{T}' = \{\tau'_1, \tau'_2, \ldots, \tau'_{B'}\}$ for updating $\eta$ ;
5      Compute the objective $J(\tau, \theta, \eta)$ over trajectories $\tau \in \mathcal{T}$ ;
6      Obtain the updated agent $\theta' \leftarrow \theta - \alpha \frac{\partial J(\tau, \theta, \eta)}{\partial \theta}$ ;
7      Compute the meta-objective $\bar{J}(\tau', \theta', \bar{\eta})$ over trajectories $\tau' \in \mathcal{T}'$ ;
8      Update agent parameter $\theta \leftarrow \theta'$ ;
9      Update meta-parameter $\eta \leftarrow \eta - \beta \frac{\partial \bar{J}(\tau', \theta', \bar{\eta})}{\partial \eta}$ ;
10 **end**

---

In the agent update and meta-parameter update steps, i.e. Line 6 and Line 9 in Algorithm 1, we can generalise the illustrated SGD updates into optimisers like RMSProp [Tieleman and Hinton, 2012] and ADAM [Kingma and Ba, 2015].

# B Detailed Hyper-Parameters used in the Atari Experiments

In Table 2, we describe the details of the important hyper-parameters used in the Atari experiments. The IMPALA hyper-parameter section is following Espeholt et al. [2018], which is provided here for self-contained purpose. The hyper-parameters in meta-gradient section are obtained by a search on six games (Beamrider, Breakout, Pong, Q*bert, Seaquest and Space Invaders) following common practice in Deep RL Atari experiments [van Hasselt et al., 2016, Mnih et al., 2016, Wang et al., 2016b]. All of the hyper-parameters are fixed across all Atari games.

| IMPALA hyper-parameter | Value |
|---|---|
| Network architecture | Deep ResNet |
| Unroll length ($n$) | 20 |
| Batch size ($B$) | 32 |
| Baseline loss scaling ($c$) | 0.5 |
| Entropy cost ($d$) | 0.01 |
| Learning rate ($\alpha$) | 0.0006 |
| RMSProp momentum | 0.0 |
| RMSProp decay | 0.99 |
| RMSProp $\epsilon$ | 0.1 |
| Clip global gradient norm | 40.0 |
| Learning rate schedule | Anneal linearly to 0 |
| Number of learners | 1 (NVIDIA P100) |
| Number of actors | 80 |
| **Meta-gradient hyper-parameter** | **Value** |
| Trace decay ($\mu$) | 0 |
| Meta learning rate ($\beta$) | 0.001 |
| Meta optimiser | ADAM [Kingma and Ba, 2015] |
| Meta batch size ($B'$) | 8 |
| Meta update frequency | Along with every agent update |
| Embedding network $\mathbf{e}_\eta$ | A linear embedding layer |
| Embedding size for $\eta$ | 16 |

Table 2: Detailed hyper-parameters for Atari experiments.

# C Implementation Details

## C.1 V-trace Return

The $\lambda$-return [Sutton, 1988] is defined as

$$G_t^\lambda = R_{t+1} + \gamma_{t+1}(1 - \lambda_{t+1})v(S_{t+1}) + \gamma_{t+1}\lambda_{t+1}G_{t+1}^\lambda \,.$$

This can be rewritten [Sutton et al., 2014] as

$$G_t^\lambda = v(S_t) + \delta_t + \gamma_{t+1}\lambda_{t+1}\delta_{t+1} + \ldots$$

$$= v(S_t) + \sum_{k=0}^{\infty} \left( \prod_{j=1}^{k} \gamma_{t+j}\lambda_{t+j} \right) \delta_{t+k},$$

where $\delta_t = R_{t+1} + \gamma_{t+1}v(S_{t+1}) - v(S_t)$, where we use the convention that $\prod_{j=1}^{0} \cdot = 1$.

This return is on-policy. For some algorithms, especially policy-gradient methods, it is important that we have an estimate for the current policy. But in the IMPALA architecture, the data may be slightly stale before the learning algorithms consumes it. Then, off-policy corrections can be applied to make the data on-policy again. In particular, IMPALA uses a *v-trace* return, defined by

$$G_t^\lambda = v(S_t) + \sum_{k=0}^{\infty} c_{t+k} \left( \prod_{j=1}^{k} \gamma_{t+j}c_{t+j} \right) \delta_{t+k} \,,$$

where $c_t = \min(1, \rho_t)$ and $\rho_t = \frac{\pi(A_t|S_t)}{\pi'(A_t|S_t)}$, where $\pi$ is the current policy and $\pi'$ is the (older) policy that was used to generate the data. Note that this return can be interpreted as an adaptive-$\lambda$ return, with a fixed adaptation scheme that depends only on the off-policy nature of the trajectory. A similar scheme was proposed by Mahmood [2017].

## C.2 Calculate the Meta-Gradient with Auto-Diff

An important fact to note in the proposed approach is that, the update rule for $\theta \rightarrow \theta'$ in first-order optimiser is linear and differentiable. In modern machine learning frameworks like TensorFlow [Abadi et al., 2016], we can alternatively obtain the meta-gradient specified in Equation (2), by utilising the automatic differentiation functionality in the framework. The only requirement is to rewrite the update operations so that the agent update can allow the gradient to flow through, noting that the build-in update operations are typically not differentiable in the common implementations.

## C.3 Data Efficiency

In order to reduce the data we needed for meta learning, we can reuse the experiences for both agent training and meta learning. For example, we can use experiences $\tau$ for updating $\theta$ into $\theta'$, validate the performance of this update via evaluating $\bar{J}$ on experiences $\tau'$. Vice versa, we can swap the roles of $\tau$ and $\tau'$, then use experiences $\tau'$ for updating $\theta$, and validate the performance of this update via evaluating $\bar{J}$ on experiences $\tau$. In this way, the proposed algorithm does not require any extra data other than the ones used to train the agent parameter $\theta$ to conduct the meta learning update on $\eta$.

## C.4 Running Speed

As for running speed, with one learner on NVIDIA P100 GPU and 80 actors on 80 CPU cores, our method runs around 13K environment steps/second, compared to around 20K environment steps/second of IMPALA baseline on the same hardware and software environments. We introduce around 35% additional compute overhead from the meta-gradient updates, however with this minor overhead we can boost the performance significantly. We'd like to highlight that the total wall clock time is about 4 hours for learning from 200 Million frames in each game.

## C.5 IMPALA

We used the IMPALA algorithm with the *deep* architecture and the *experts* mode of training. In this mode, a separate agent is trained on each environment (i.e. the standard RL setting), as opposed to a multi-task setting. Population-based training was not utilised by the IMPALA experts in Espeholt et al. [2018]; we follow this convention. In principle, $\gamma$ and $\lambda$ could be exposed to population-based training (PBT) [Jaderberg et al., 2017a], however, this would blow up the computation time by the size of the population (24 in Espeholt et al. [2018]), which is beyond reach for typical experiments; furthermore adaptation by PBT does not exploit the gradient and is therefore perhaps less likely scale to larger meta-parameterisations.

## D Baseline Experiments on Atari Games

In this section, we show the baseline experiment results on Atari, including grid search and an approach to predict auxiliary value functions for multiple discount factors.

### D.1 Results of Grid Search of Discount Factor $\gamma$ on Atari Experiments

We conduct simple grid search on the discount factor $\gamma$, i.e., let $\gamma = 0.99, 0.995, 0.998, 0.999$ respectively, and apply it in the IMPALA framework. The grid search of discount factor $\gamma$ is to find some good $\bar{\gamma}$ ($\bar{\eta} = \bar{\gamma}$ in this case) to be used in the meta-objective $\bar{J}(\tau', \theta', \bar{\eta})$, so that the meta learning approach can have a good proxy to the true return to learn from.

Table 3: Performance comparison of IMPALA baseline with different discount factor $\gamma$, all scores are human-normalised [Mnih et al., 2015, Wang et al., 2016b, van Hasselt et al., 2016].

|  | Human starts | No-ops starts |
|---|---|---|
| Agents | median | median |
| $\gamma = 0.99$ [Espeholt et al., 2018] | 144.4% | 191.8% |
| $\gamma = 0.995$ | 211.9% | 257.1% |
| $\gamma = 0.998$ | 208.5% | 210.7% |
| $\gamma = 0.999$ | 114.9% | 153.0% |

As we can see from Table 3, the discount factor $\gamma$ has huge impact on the agent performance.

In addition, we use grid search as a way to pick the best $\gamma$ for each game and evaluate each game with the chosen $\gamma$ in a fresh run. The grid search is performed over the same range of discount factors as above. This obtains a median score of 214.7% under human-starts evaluation, compared to 268% with our meta-gradient method.

### D.2 Results of Learning Auxiliary Value Functions for Multiple Discount Factors

Since the agent is learning with an online adapting discount factor $\gamma$, one hypothesis of the performance improvement could be, the agent is learning from multiple discount factors and predicting the value functions over multiple timescales. To validate the hypothesis, we conduct a baseline experiment, which performs auxiliary tasks to predict the value functions with additional discount factors (e.g. the agent augments IMPALA ($\gamma = 0.99$) with auxiliary tasks to predict value functions for $\gamma = \{0.995, 0.998, 0.999\}$). The additional value predictions share the representation of the agent and have a linear layer to predict the values for each of the additional discount factor. We compare the results against our meta-gradient approach and the IMPALA baseline ($\gamma = 0.99$). We report the median human-normalised scores under human-start evaluation condition, which the auxiliary value predictions baseline obtains 152.5%. Though it improves from the IMPALA baseline (144.4%), it's much lower than what the meta-gradient approach achieves (233.2%), suggesting the primary benefit indeed comes from adapting the meta-parameter $\eta$.

# E   Additional Experiment Results on Atari Games

## E.1   Relative Performance Improvement in Individual Games

In this section, we provide the relative performance improvement of the meta-gradient algorithm compared to the IMPALA baselines in individual Atari 2600 games. We show the results of adapting discount factor, i.e., $\eta = \{\gamma\}$, in Figure 2, and results of adapting both discount factor and bootstrapping parameter, i.e., $\eta = \{\gamma, \lambda\}$, in Figure 3.

Figure 2: The relative performance improvement of the meta-gradient algorithm, adapting discount factor, i.e., $\eta = \{\gamma\}$ with $\bar{\gamma} = 0.99$, compared to the baseline IMPALA ($\gamma = 0.99$) in individual Atari 2600 games, where the gain is given by $\frac{\text{proposed}-\text{baseline}}{\max(\text{human},\text{baseline})-\text{random}}$ [Wang et al., 2016b]. Improvement over 200% is capped into 200% for visualisation.

Figure 3: The relative performance improvement of the meta-gradient algorithm, adapting both discount factor and bootstrapping parameter, $\eta = \{\gamma, \lambda\}$ with $\bar{\gamma} = 0.995$ and $\bar{\lambda} = 1$, compared to the baseline IMPALA ($\gamma = 0.995$) in individual Atari 2600 games, where the gain is given by $\frac{\text{proposed}-\text{baseline}}{\max(\text{human,baseline})-\text{random}}$ [Wang et al., 2016b]. Improvement over 200% is capped into 200% for visualisation.

## E.2 Training Curves

In this section, we provide the training curves for two representative experiments: adapting $\eta = \{\gamma\}$ with $\bar{\gamma} = 0.99$ (Figure 4) and adapting $\eta = \{\gamma, \lambda\}$ with $\bar{\gamma} = 0.995$ (Figure 5).

Figure 4: Training curves for meta learning $\eta = \{\gamma\}$ with $\bar{\gamma} = 0.99$. We provide the comparison of scores against baseline, and the change of $\gamma$ for each game. Best viewed in electronic version.

Figure 5: Training curves for meta learning $\eta = \{\gamma, \lambda\}$ with $\bar{\gamma} = 0.995$ and $\bar{\lambda} = 1$. We provide the comparison of scores against baseline, the change of $\gamma$, and the change of $\lambda$ for each game. Best viewed in electronic version.