[Reviews · NeurIPS 2018]

Reviewer 1



In this paper, the authors propose a meta-learning style algorithm in order to optimize the design parameter of RL methods, in an interesting online fashion, while the RL agent is training. In RL, it is always a been a question about what is a right discount factor \gamma or if TD(\lambda) method is used, what is the best \lambda. In this paper, the authors provide a method such that the agent( or the meta-learner) figures out itself what is the right choice of parameters to use at the time. For a given objective function J(\theta, \gamma, \lambda), the RL agent computes the gradient with respect to \theta (the policy (and/or value function) parameter) in order to learn a better policy ( and/or a better value function), then update the model (policy/value) parameters. Then if the agent's desire is to maximize the return under a specific \lambda' and \gamma', it chooses the \lambda and \gamma such that the agent intrinsically maximizes the return under \lambda' and \gamma' while the gradient of J(\theta, \gamma, \lambda) with respect to \theta is considered. ***************** There are two comments which they did not affect the scoring but are needed to be addressed. 1) The first line of the abstract is not true in general. I would suggest that the authors restate it. 2) Cite and mention this work "On Structural Properties of MDPs that Bound Loss due to Shallow Planning" which study why you even do not want to use the given discount factor and why you would even rather choose a smaller one. ***************** The paper is clearly written, also deliver the idea and method clearly. There are two major issues. 1) In equation 11 and 13, when J' objective, the square error, is used to optimize for \gamma and \lambda, the author should consider that minimizing J' also minimizes the variance of g which means that the suggested \gamma and \lambda also contribute to minimizing the variance of g which is not the desire of the proposed optimization \citet{Learning near-optimal policies with Bellman-residual minimization based fitted policy iteration and a single sample path} 2) Despite the interesting idea, the empirical results do not provide much beyond the modest performance of impala.

Reviewer 2



A gradient-based hyper-parameter technique for reinforcement learning is proposed, and implemented on top of the IMPALA architecture. The algorithm consists in estimating the gradient of an objective J’ w.r.t. hyper-parameters eta, using the chain rule: d_J’/d_eta = d_J’ / d_theta’ * d_theta’ / d_eta, where theta’ = theta + f(tau, theta, eta) is the updated vector of model parameters after observing trajectory tau with current parameters theta and hyper-parameters eta. In other words, we look for a small change in hyper-parameters that would improve the performance of the agent after a parameter update, as measured by the objective J’. This objective is based on independently sampled trajectories tau’ and a fixed set of hyper-parameters eta’. In order to accommodate the fluctuations in hyper-parameters eta, they are provided as input to the actor and critic networks. The main experiments are on Atari, where eta = {gamma, lambda} (respectively the discount factor and the bootstrapping parameter in TD(lambda)). They show a significant improvement over the standard IMPALA architecture (which uses the best gamma = 0.995 and lambda = 1 found by grid search), in terms of median human-normalized score. There is no doubt this is an interesting paper. Just the fact that the discount factor matters so much (huge improvement with vanilla IMPALA when going from 0.99 to 0.995, but poor results at 0.999) is quite notable to me: to my knowledge it is not something that had been investigated previously (on Atari specifically). I actually find this somewhat worrying, since it sheds doubts on the validity of the standard practice of training with gamma = 0.99 then evaluating on the total accumulated reward for benchmarking purpose, as is done in almost all deep RL papers evaluated on Atari... but this is another discussion. The core empirical results are convincing enough: there is a relatively consistent improvement over the baseline IMPALA, which is a solid benchmark. It is still pretty far from Ape-X DQN but since the number of environment steps is much lower, this may not be a fair comparison to make. As shown in Fig. 2 of the Appendix, though, there are some games where the proposed technique clearly hurts: it would have been interesting to try and understand what is going on in these games that makes the hyper-parameter adaptation scheme fail. In addition, I believe there should have been some comparison to other hyper-parameter adaptation techniques, since the ability to improve on a fixed gamma over all games is not particularly surprising (given the highly varying nature of Atari games). At the very least, a comparison to grid search should have been done where one would pick the best gamma *for each game*. I would also argue that the proposed method is not just a hyper-parameter adaptation scheme: since hyper-parameters eta are provided as input to the actor and critic, the model is essentially performing multi-task learning (learning to predict returns at multiple timescales). It has been observed before that additional tasks can have a beneficial effect (ex: the UNREAL agent), and in my opinion this begs the question as to whether the improvement comes mostly from changing eta or from learning to predict over multiple eta’s. Note that even if eta eventually converges to a fixed value, the optimization w.r.t. J’ encourages updates that give good results for eta’ != eta, so the model is still learning with at least two time scales. Related to this last point, the motivation given by the authors for adding eta as a model input is that “the return function gη(τ ) is non-stationary”, but it is not clear to me whether this is the main one, since eta does not change that fast. Another clearly important factor, not mentioned in the paper, is that it allows to compute J’ on eta’ != eta, while otherwise it would seem difficult to optimize J’ with the “wrong” hyper-parameters (by the way please mention 1.4 earlier, or add a reference to it in 1.2, because eq. 11 got me puzzled for a while due not being aware of that yet). To summarize, my point is that even if there is no doubt the method works, I wish there had been more efforts to understand why. In this respect, the illustrative examples from Section 2 could have help shed light on the inner workings of the algorithm. Unfortunately I found them extremely confusing. I suspect that my understanding of “state-dependent” hyper-parameter may not match the actual implementation. For instance when I think of a state-dependent gamma, to me it means that the Bellman equation would be written V(s) = E[R(s)] + gamma_s V(s’), where gamma_s is the state-dependent discount factor. But then it makes no sense for gamma_s to be near zero in “noise” states in Fig. 1a, as it would mean we only learn V(s) from the noise R(s), which can’t be right. The algorithm also learned a specific high value for gamma_9 while gamma should not matter in that state, since the next state is terminal and thus V(s’) = 0. The second example (Fig. 1b) is equally confusing, if not more. Since even-numbered states are aliased, the agent sees them as a single state. That state is visited as often as a bottleneck state (contradicting “The fully observed bottleneck states are visited frequently”), and has essentially the same properties regarding the noise in the (immediate) reward distribution: a random (due to aliasing) number chosen among -r1, .. -rn (vs. among r1, ..., rn in a bottleneck state). I may have misunderstood the meaning of “aliased” states, in which case a clarification would be appreciated. In any case, I was unable to derive a better understanding of the proposed method from these two toy examples. There are other unclear parts in the paper, even if the high-level idea is easy to grasp. One of them is how the dependence of the actor pi and critic v on the hyper-parameters eta is handled w.r.t. gradient computations: the derivations in 1.2 and 1.3 assume that v and pi do not depend on eta. But then 1.4 adds such a dependency... Are you actually backpropagating through the whole computational graph, or “cutting” this specific part of the gradient? (and in the former case, does that include backpropagating through the IMPALA importance sampling weights?). I believe this is an important point to clarify and motivate. Other unclear points include: - The inclusion of d_thera / d_eta in eq. 3 (first term). I do not understand what it is supposed to represent. From my point of view, when starting from theta, modifying eta has no effect on theta itself: it can only change future values theta’. The “accumulative trace” approximation looks fishy to me as I fail to see what it is meant to approximate: I see it simply as a momentum applied to d_f / d_eta. Note also that when reading the text it is not obvious that this is a recursive update with z’ being the new value of z used in the nest iteration. - The motivation for keeping only the policy gradient objective when defining J’ in eq. 14. It is not obvious to me that this is the best choice, since a better critic v could also have a positive impact on the agent’s performance. - The motivation for using gamma’ = 0.995 in Atari experiments: should not it be chosen based on the performance of the adaptive algorithm, rather than the best fixed gamma? - The exact meaning of batches (“meta batch size” in Table 2): is it the case that J’ is actually evaluated over a mini-batch of trajectories, instead of a single trajectory tau’ as the equations suggest? - More generally, adding the pseudo-code of the algorithm in the Appendix would be very helpful, especially with 1.5 showing there are important implementation “details” Minor points (missing from original review, but thankfully the character limit is gone): - I would suggest to remove eta from the definition of n-step returns in eq. 6 since the proposed method can’t optimize discrete hyper-parameters (a limitation, by the way, which is not clearly stated in the abstract & conclusion) - In eq. 8 dividing J by two would slightly simplify subsequent equations - Caption of Fig. 1: “the first one shows how the meta-parameter γ or λ adapts” => please specify “for each state” - What is the value of n in experiments? Is it the “unroll length” 20? - l. 177 please refer to Fig. 1 (top), as trying to understand the examples from their text description is difficult - Please specify the values of respectively lambda and gamma used in the two illustrative examples - l. 202 why not write eta = {gamma, lambda} ? - In Atari do you use lambda’ = 1? If yes how do you deal with the fact a sigmoid can’t be equal to 1 (footnote 2) and you need the logit as input to the actor and critic to compute J’? (the same question actually arises for the illustrative examples) - In Appendix Fig. 2 and 3 please specify eta’ Regarding the author response: First, thank you for the detailed response. My main concerns have been addressed by the extra results related to the grid search on gamma and the multi-task experiment, so I've changed by recommendation to accept. A few follow-up remarks that hopefully you can take into account when preparing the final version: 1. You motivated including d_theta / d_eta in eq. 3 with a recurrent view of theta as a function of eta (response l. 35). However, in practice eta has not been kept fixed during the previous steps (since the algorithm is adapting it), and thus theta is not just a function of the current eta, but depends of the value of eta at each previous timestep. As a result this derivative still makes no sense to me (NB: and even if eta had been kept fixed, I am still not sure it would be a good idea to follow this gradient, since we're never going back in time to "improve the past" -- changing eta only affects future updates) 2. In your first toy example (Fig. 1a) you say that (response l. 27) “in noisy states setting e.g. gamma2 ≈ 0 avoids receiving the "noise" reward”. However this effectively shuts down all subsequent rewards as well when e.g. computing the value function for the very first state. I don’t see how one can properly estimate the long-term value v_1 of the first state (with gamma’ = 1) if gamma_2 = 0 since the only return the agent will see from state 1 is 0.1. 3. Your second example (Fig. 1b) finally "clicked" for me when I wrote down what the return would look like depending on when to bootstrap (with binary lambda's). For instance using as target either r_i + v(aliased_state) or r_i - r_i + v(bottleneck_state). It becomes clear that the second one is less noisy since the two consecutive rewards immediately cancel each other in the return. So maybe you could write it down explicitly in the paper as well (possibly in the Appendix if not enough room). 4. Regarding point R2.7 in your response, I actually didn't think of using gamma'_t = gamma_t. I had in mind a simpler "classical" hyper-parameter selection strategy, e.g. doing a grid-search over gamma'. I don't disagree that in the absence of other any information, using gamma' = the best fixed gamma seems like a sensible solution, however it remains a somewhat arbitrary choice that was not motivated in the submission. Thanks! ======== One last thing, regarding point 1: I posted the following on the reviewers’ discussion board, to try and explain my point of view to fellow reviewers (none of them could understand the role of mu by the way, so I really hope this part gets fixed in a final version): I see the meta-optimization problem as an iterative procedure, where at each step we start from the previous theta and want the update f (as in eq. 1) to improve on the meta-objective J'. We can control f through hyper-parameters eta, and eq. 2 tells us the direction to follow in the hyper-parameter space. This direction requires computing d_theta' / d_eta, which from eq. 1 is *equal* to d_f / d_eta, because theta is the (fixed) starting point at the current step, and is not affected by any subsequent change to eta. Plugging this in eq. 2, we get that d_J' / d_eta = d_J' / d_theta' * d_f / d_eta, which is indeed the update direction that is being used when when mu = 0 (see eq. 5 where z' = d_f / d_eta). So, to me, mu=0 is not an approximation, it yields the correct update direction for the optimization objective. Using mu > 0 amounts to adding momentum to the gradient d_f / d_eta, which I guess could be argued to stabilize training in some situations. Note that it's not exactly similar to regular momentum in SGD because this momentum is not applied to the final update direction (eq. 2), but only to one of its components. So it's not really clear what's going to happen with mu > 0, and since all experiments reported in the submission are with mu=0, there's no way to tell how sensitive performance is to changes to this parameter. That being said, since from my point of view mu=0 is the natural thing to do, I would not reject the paper based only on its lack of clarity regarding mu's role.

Reviewer 3



Summary: This paper introduces a gradient-based meta-learning method that learns to tune the hyper-parameters for TD ($\lambda$) with discount factor $\gamma$ and bootstrapping $\lambda$ in order to optimize the estimation of return. The meta-learning algorithm is online and able to adapt to the non-stationarity of return function from interacting with the environment. Extensive results on Atari games demonstrate the effectiveness of the proposed meta-learning algorithm. This paper looks very interesting and I vote for the acceptance. Quality: This work is of high quality. This meta-RL is based on online cross-validation: training with update rule function and evaluating the performance of the updated new parameter with successive experience samples alternatively. This paper introduces the the hyper-parameter of return function $\eta$ as the meta-parameter, incorporating it into update rule function $f(\tau, \theta, \eta)$. The update rule function $f$ is differential and is used to update the parameter of interest $\theta$, e.g. for value function or policy. Therefore, we can learn the new parameter $\theta'$ by the update rule. On the other hand, this paper introduces the evaluation function $J'(\theta', \tau', \eta')$ to measure the performance of new updated parameter $\theta'$ on an independent sample $\tau'$ - this evaluation is denoted as meta-objective. This meta-objective is further used to update the meta-parameter $\eta$ in the descent direction of the meta-gradient. The trick here is that to find a reasonable return proxy for the performance evaluation, we still need to set the hyper-parameter $\eta' = \{\gamma', \lambda' \}'$ manually, which is sensitive to the return function. Furthermore, additional hyper-parameters such as the learning rate is introduced to learn the meta-parameter. To make the value function and policy adaptive to the non-stationary meta-parameter throughout the training process, the authors propose to argument the meta-parameter as the input 'state' of the value function and policy. Clarity: The paper is generally well-written and structured clearly and I enjoyed reading it. This paper is easy to understand even for those with only a basic background in machine learning. One minor comment is for Eq. (12) between line 128 and 129: it might be better to provide the objective function $J(\tau, \theta, \eta)$ for a clear understanding of the gradient w.r.t. $\theta$ (Eq.12). Originality: The idea of meta-RL, or more specifically, gradient-based meta-learning adapting to the environment, is natural and no longer new. The authors cite the relevant papers, e.g. Finn et. al. Model-Agnostic Meta-Learning for Fast Adaptation of Deep Networks (ICML 17), Al-Shedivat et. al. Continuous Adaptation via Meta-Learning in Nonstationary and Competitive Environments (ICLR 18). Recently, Xu et. al. Learning to Explore via Meta-Policy Gradient (ICML 18) is a new, gradient-based, meta-RL method that learns to do exploration. However, adopting the meta-RL method to learn the return function adaptively is certainly novel and interesting. Significance: It is important to estimate and optimize the non-stationary return function in Reinforcement Learning by trading-off the bias and variance. This paper provides an interesting perspective from meta-RL and is able to adjust the sensitive hyper-parameters automatically to achieve better results. -------- After Rebuttal ------- After reading others' reviews, discussions and authors' feedback, I am a little bit concerned about the approximation ($\mu=0$) in Eq.4. Very curious to see what is the $\mu$'s impact empirically and what is the reasoning? Is $\mu$ sensitive to the performance?

Reviewer 4



This paper proposes meta-learning the hyperparameters associated with policy gradients, using the fact that such hyperparameters interact with the learner in a differentiable way. This work’s motivation is clear (illustrated with examples section 2). The derivations are clear (a straightforward application of “backprop through learning”) and experiments are impressive. I am, however, not completely up-to-date with research in meta reinforcement learning and thus not absolutely certain how novel this method is. Overall, I think this is a solid work that addresses the problems associated with policy gradient methods using ideas from meta-learning.